# Security Risk Level Prediction of Carbofuran Pesticide Residues in Chinese Vegetables Based on Deep Learning

**DOI:** 10.3390/foods11071061

**Published:** 2022-04-06

**Authors:** Tongqiang Jiang, Tianqi Liu, Wei Dong, Yingjie Liu, Qingchuan Zhang

**Affiliations:** 1National Engineering Research Centre for Agri-Product Quality Traceability, Beijing Technology and Business University, Beijing 100048, China; jiangtq@btbu.edu.cn (T.J.); 2031101006@st.btbu.edu.cn (T.L.); liurita0911@163.com (Y.L.); 2School of E-Business and Logistics, Beijing Technology and Business University, Beijing 100048, China

**Keywords:** security risk assessment, carbofuran, vegetables, security risk level prediction, deep learning

## Abstract

The supervision of security risk level of carbofuran pesticide residues can guarantee the food quality and security of residents effectively. In order to predict the potential key risk vegetables and regions, this paper constructs a security risk assessment model, combined with the k-means++ algorithm, to establish the risk security level. Then the evaluation index value of the security risk model is predicted to determine the security risk level based on the deep learning model. The model consists of a convolutional neural network (CNN) and a long short-term memory network (LSTM) optimized by an arithmetic optimization algorithm (AOA), namely, CNN-AOA-LSTM. In this paper, a comparative experiment is conducted on a small sample data set of independently constructed security risk assessment indicators. Experimental results show that the accuracy of the CNN-AOA-LSTM prediction model based on attention mechanism is 6.12% to 18.99% higher than several commonly used deep neural network models (gated recurrent unit, LSTM, and recurrent neural networks). The prediction model proposed in this paper provides scientific reference to establish the priority order of supervision, and provides forward-looking supervision for the government.

## 1. Introduction

With the improvement of people’s living standards, consumers pay more and more attention to the quality and security of food. As an indispensable food in people’s diet, the quality and security of vegetables is directly related to human health, and the security risk level of dietary intake of pesticide residues is an important performance to measure the quality and security of vegetables. As a common pesticide, carbofuran is a broad-spectrum, efficient, low residue, high toxicity insecticide, acaricide, and nematocide. It is easy to use, labor-saving, cheap, and can be applied to cotton, rice, sugar cane, peanut, corn, sugar beet and another 80 crops for pest control. However, its toxicity is high, and acute exposure can inhibit the cholinesterase activity in the body, leading to tears, vomiting, salivation, pupil narrowing, spasm, and serious cases of blood pressure drop or unconsciousness, but also has reproductive toxicity, developmental toxicity, gene toxicity, neurotoxicity and so on [1]. In addition, studies have shown that carbofuran has the effect of environmental hormones [2], which can act on the nucleus and cause genetic variation [3]. In April 2002, the Ministry of Agriculture of China proposed to stop accepting new registration of carbofuran, stop approving sub-packaging registration of highly toxic pesticides, and cancel the registration of carbofuran on citrus trees. In June 2002, the Ministry of Agriculture also announced that carbofuran should not be used or only have restricted use in vegetables, fruit trees, tea leaves and Chinese herbal medicines. However, due to its high efficiency and low cost, there are still reports of excessive levels of carbofuran in vegetables and fruits in many provinces [4,5,6].

Frequent food security incidents and increasingly stringent food security regulations in international trade have put forward higher requirements for food quality and security supervision [7]. Through food security risk assessment and early warning analysis, on the one hand, it can provide data support and supervision basis for market regulation, and on the other hand, it can provide reference for consumers to buy vegetables.

He et al. [8] measured the residues of 37 pesticides in Xinjiang jujube, assessed the risk of chronic dietary intake and acute dietary intake, and used the risk matrix to rank the risk of pesticide residue. Zhao et al. [9] studied the influence of cooking styles in southern China on the actual intake and long-term exposure risk of pesticide residues in vegetables, and analyzed the harm level of pesticide residues in four cooking styles to the human body. Fang et al. [10] evaluated the ingestion risk of pesticide residue in celery diet in China, and ranked the ingestion risk of each tested pesticide according to a pre-defined ranking matrix. D. et al. [11] evaluated the health risks of pesticide residues in fruits and vegetables in farms and markets in the West Indian Himalayas. Based on the prediction of chronic health hazards from estimated daily intake, organophosphorus were found to pose a health threat to children in the study area. Most of the existing literature has evaluated and graded individual vegetables or fruits, and most of the studies were conducted in different regions. There is little research about comprehensive security risk assessment and grading of pesticide residues in various vegetables in many provinces in China.

In risk prediction, deep learning methods such as long short-term memory network (LSTM), gated recurrent unit (GRU) and recurrent neural networks (RNN) can capture high-dimensional features and show temporal dynamic behavior appropriately [12]. These methods have been used in weather forecast or travel time prediction, achieving higher accuracy prediction [13,14]. Time series analysis can analyze the historical data of dynamic systems and predict the future operation mode [15]; therefore, this feature also meets the requirements of food security risk prediction. Yu et al. [16] used the autoregressive integrated moving average model (ARIMA) to predict pesticide residues in two vegetables in different seasons in Sichuan, China. Xin et al. [17] predicted the heavy metal content in lettuce leaves by deep learning combined with hyperspectral imaging technology.

This paper establishes a deep learning-based prediction model of security risk level of carbofuran pesticide residue in Chinese vegetables by analyzing 200,000 measurements of national monitoring sampling data of carbofuran pesticide residue contamination in vegetables in 2019. Firstly, the food security risk evaluation model is constructed. Based on the national sampling data of carbofuran in vegetables in 2019 and the weekly consumption data of vegetables in each province, the weekly index value of the evaluation model of each vegetable in each province is calculated. Then, the national sampling data are clustered by the k-means++ algorithm [18] to classify the security risk level based on the indexes calculated. Finally, based on the deep learning model, the historical data are used to predict the security risk evaluation indexes of vegetables in each province, and the distance between the predicted security risk evaluation indexes and the security risk level clustering center is measured to determine the security risk level. The model proposed in this paper quantifies the sampling data into security risk levels and predicts the evaluation indexes, which provides a scientific reference for establishing the priority of supervision and provides a forward-looking basis for government supervision.

## 2. Materials and Methods

### 2.1. Materials

#### 2.1.1. Data Sources

The data in this study comes from the national food security sampling data in 2019, covering 20 provinces. The vegetables are divided into eight categories, including legumes vegetables, leafy vegetables, solanaceous vegetables, bulb vegetables, melon vegetables, roots and potato vegetables (except potatoes), brassica vegetables and potatoes, with a total of 13,222 samples. According to the maximum residue limit for pesticide from the Food of National Food Security Standard, the content of carbofuran is determined. China’s national food security standards set the limit of carbofuran in legumes vegetables, leafy vegetables, solanaceous vegetables, bulb vegetables, melon vegetables, roots and potato vegetables (except potatoes), brassica vegetables and other vegetables as 0.02 mg/kg, and the limit of carbofuran for potatoes is 0.1 mg/kg, higher than the limit of other vegetables.

The vegetable data of residents’ consumption comes from the fifth Total Diet Study [19]. This survey adopts stratified and multi-stage cluster random sampling method proportional to the population to carry out a dietary questionnaire survey on the main food consumption of residents in 20 provinces of China.

#### 2.1.2. Data Preprocessing

According to the principle of Credible Assessment of Low-level Contaminants in Food, proposed at the second meeting of WHO Global Environmental Monitoring System/Food Contamination Monitoring and Assessment Programme (GEMS/FOOD), when the proportion of undetected data is less than 60%, all undetected data are replaced by 1/2 of the detection limit (LOD), and when the proportion of undetected data is higher than 60%, all undetected data are replaced by LOD [20]. As the undetected data of carbofuran in this study is far less than 60%, 1/2 LOD value is assigned to all undetected data in this study for statistical calculation.

### 2.2. Security Risk Assessment Model

According to the security risk assessment method and the purpose of setting the model, based on the main influencing factors of health risks caused by food contaminants, the Nemerow integrated pollution index (NIPI), the acute exposure assessment (AEA), and the chronic dietary exposure assessment (CDEA) are used as the three indexes of the security risk assessment model. The median (P50) and 97.5 quantile (P97.5) of carbofuran in vegetables are used to calculate the exposure of carbofuran at different pollution levels.

#### 2.2.1. Nemerow Integrated Pollution Index

The Nemerow integrated pollution index can reflect the characteristics of food pollution, and it is used to evaluate the pollution of air [21], heavy metals in soil [22,23], rice [24] and vegetables [25,26,27] by researchers. In this paper, the Nemerow integrated pollution index is used to calculate the pollution degree of carbofuran from sampling samples based on sampling data of each province, and the expression is as follows:(1)$$P_{i,j}=\frac{X_{i,j}}{S_{j}}$$
where, $P_{i,j}$ is the pollution index of vegetable *j* in province *i*; $X_{i,j}$ is the detected value (mg/kg) of carbofuranin vegetable *j* in province *i*; and $S_{j}$ is the national limit standard of carbofuran in vegetable *j* (mg/kg).
(2)$$P_{c\left(i.j\right)}=\sqrt{\frac{P_{\max(i,j)}^{2}+P_{ave\left(i,j\right)}^{2}}{2}}$$
where, $P_{c(i,j)}$ is the Nemerow integrated pollution index of vegetable *j* in province *i*; $P_{\max(i,j)}$ is the maximum pollution index of vegetable *j* in province *i*; and $P_{ave(i,j)}$ is the mean value of pollution index of vegetable *j* in province *i*.

#### 2.2.2. Acute Exposure Assessment

Acute exposure assessment of food is widely used as an index of the acute impact of agricultural and veterinary drug residues [28,29,30] and microorganisms on human health. This index is to evaluate the exposure amount of a certain substance consumed by diet within 24 h. In this paper, the point assessment method is used to calculate acute exposure assessment. The expression is as follows:(3)$$EDI_{97.5\left(i,j\right)}=\frac{F_{97.5\left(i,j\right)}\times C_{\max\left(i,j\right)}}{W}$$
where $EDI_{97.5\left(i,j\right)}$ is the 97.5 percentile of daily intake of carbofuran from vegetable *j* per kilogram of body weight in province *i* (mg/kg bw), $F_{97.5\left(i,j\right)}$ is the 97.5 percentile (P97.5) of consumption of vegetable *j* in province *i* (kg/d), $C_{\max\left(i,j\right)}$ is the maximum detected value of carbofuran of vegetable *j* in province *i* (mg/kg), and *W* is the average body weight of residents (60 kg).

#### 2.2.3. Chronic Dietary Exposure Assessment

The chronic dietary exposure assessment represents the risk of chronic dietary intake of carbofuran based on the average daily intake of carbofuran per kilogram of body weight in the population. The expression is as follows:(4)$$EDI_{50\left(i,j\right)}=\overset{j}{\underset{j^{\prime }}{\sum }}\frac{F_{i,j^{\prime }}\times C_{i,j^{\prime }}}{W}$$
where $EDI_{50\left(i,j\right)}$ is the average daily intake of carbofuran from vegetable *j* per kg of body weight in province *i* (mg/kg bw), $F_{i,j^{\prime }}$ is the average consumption of fine vegetable $j^{\prime }$ in province *i* (kg/d), $C_{i,j^{\prime }}$ is the average content of carbofuran in fine vegetable $j^{\prime }$ in province *i* (mg/kg), and *W* is the average body mass of residents (60 kg).

### 2.3. Security Risk Classification Based on K-Means++

The security risk assessment model of carbofuran in vegetables is established by integrating the Nemerow integrated pollution index, the acute exposure assessment, and the chronic dietary exposure assessment. Based on the security risk assessment model, food security risks are classified. In this paper, clustering algorithm is used to select the optimal security risk level division to reduce the influence of subjective factors. Cluster algorithm is a process of dividing a given sample into multiple clusters to mine the deep information of data. Its goal is to make the samples in the same cluster have high similarity, and the samples in different clusters have low similarity. The k-means++ algorithm has good selection of initial sample points, good support for high-dimensional data, and can achieve good clustering performance in the arbitrary shape of sample space, which is suitable for analyzing model data of this study.

The advantage of k-means++ is that there is no need to artificially determine the initial clustering centers. The basic idea of selecting initial seeds in this algorithm is that the distance between the initial clustering centers should be as far as possible, and the specific process is as follows:(1)Select a point randomly from the set of input data points as the first clustering center.(2)For each point x in the data set, calculate the distance D(x) between it and the nearest cluster center (referring to the existing cluster center).(3)A new data point is selected as the new clustering center, and the selection principle is as follows: the point with larger D(x) has a higher probability of being selected as the clustering center.(4)Repeat (2) and (3) until k cluster centers are selected.(5)The k initial clustering centers are used to run the standard k-means algorithm.

### 2.4. CNN-AOA-LSTM Security Risk Level Prediction Model Based on Attention Mechanism

#### 2.4.1. Framework of CNN-AOA-LSTM Model

This paper proposes a CNN-AOA-LSTM security risk level prediction model based on attention mechanism [31], as shown in Figure 1. Security risk assessment indexes for each vegetable in each province are predicted, which are used to calculate the security risk level. The model is divided into five layers: data layer, CNN layer, attention mechanism layer, AOA-LSTM layer, and prediction security risk level layer. The structure of the neural network model is shown in Figure 2.

Firstly, CNN is used to extract the features of historical food security risk assessment index data. The convolutional layer expands the depth, the pooling layer reduces the number of parameters by dimensionality reduction, and the fully connected layer transforms the features into one-dimensional vectors to complete feature extraction.

Secondly, this paper introduces the attention mechanism to allocate the probability weight to enhance the proportion of useful information. Due to the drug properties of carbofuran, the PH of soil has a certain influence on its absorption and decomposition. On the other hand, Deutsch et al. [32] has shown that higher temperatures produce more crop-eating pests. Hence, vegetable farmers often spray more pesticides, especially in the summer. Moreover, the interval between spraying time and harvest time is not fixed. If the degradation of pesticides is not enough, the levels of pesticide residues will be higher. Therefore, this paper introduces Chinese soil PH value and the air temperature as input of attention layer.

Thirdly, since this paper is a small sample data set, the selection of hyperparameters is particularly important. The feature matrix introducing attention is input into the AOA-LSTM neural network to learn the change rule of security risk assessment index data to predict future security risk assessment indexes, and the LSTM hyperparameter selection is optimized through AOA [33], so as to improve the prediction accuracy.

L. et al. [33] proposed that model optimization was carried out by using the definition of the four operators of addition, subtraction, multiplication and division in mathematics. The error of LSTM network is a fitness function, and the role of AOA is to find a group of optimal hyperparameters to minimize the network error. In the algorithmic exploration phase, multiplication strategy and division strategy are used for global search, which can improve the dispersion of solutions, enhance the global search and overcome premature convergence, and realize global search. In the algorithm development phase, the addition and subtraction strategies are used to reduce the dispersion of the solutions, which is conducive to the full development of the population in the local range and strengthen the local optimization ability of the algorithm.

Finally, the predicted security risk assessment indexes are output by the output layer, which are used to measure the nearest distance between the comprehensive assessment indexes and the cluster centers, and to classify security risk levels.

#### 2.4.2. Attention Mechanism Based on PH of Soil and Temperature

China has a large amount of land, and the PH value of land in different regions varies greatly [34]. Moreover, the PH of the same mineral soil also changes slightly over time [35], and the reason is that the PH decreases during growing season of the crop as a result of acid production by microorganisms and higher plant roots. In addition, it has slight influence on soil PH by some conditions, such as rising and falling soil moisture, which causes salt to move in and out of the soil layer, affecting soil PH; therefore, rainfall can affect the PH of soil. Xie et al. [35] has concluded that the disappearance rate of Carbofuran is positively correlated with the PH of soil.

In addition, Deutsch et al. [32] has found that as temperatures rise, almost all insects speed up their reproduction and metabolism, with disastrous consequences for the world’s food supply.

The national meteorological data of each province in 2019 can be obtained from the China Meteorological Science Data Sharing Service network, including temperature and rainfall.

According to Chinese soil PH distribution, rainfall and expert consultation, soil PH $p_{m}^{i,j}$ on day j of the week i for each province is added into the matrix $P_{m}$, where m is one of the 20 provinces mentioned above, i is week i, and j is day j of week i (j=0,1,…,6). The annual soil PH distribution matrix is obtained in Formula (5). Each value in the matrix is set to reciprocal and normalization, and we get $\tilde{P_{m}}$.
(5)$$P_{m}=\left[\begin{array}{c} p_{m}^{0,0},\ldots ,p_{m}^{0,j},\ldots ,p_{m}^{0,6}\\ \vdots \\ p_{m}^{i,0},\ldots ,p_{m}^{i,j},\ldots ,p_{m}^{i,6}\\ \vdots \\ p_{m}^{53,0},\ldots ,p_{m}^{53,j},\ldots ,p_{m}^{53,6} \end{array}\right]$$


(6)
$$\tilde{P_{m}}=\left[\begin{array}{c} \tilde{p_{m}^{0,0}},\ldots ,\tilde{p_{m}^{0,j}},\ldots ,\tilde{p_{m}^{0,6}}\\ \vdots \\ \tilde{p_{m}^{i,0}},\ldots ,\tilde{p_{m}^{i,j}},\ldots ,\tilde{p_{m}^{i,6}}\\ \vdots \\ \vdots \tilde{p_{m}^{53,0}},\ldots ,\tilde{p_{m}^{53,j}},\ldots ,\tilde{p_{m}^{53,6}} \end{array}\right]$$


In terms of the national meteorological data in 2019, the temperature $t_{m}^{i,j}$ on day j of the week i for each province is added into the matrix $T_{m}$, and the letters have the same meaning as above. Finally, we get the national temperature distribution matrix as in Formula (7). Then each value in the matrix is set to normalization, and we get $\tilde{T_{m}}$.
(7)$$T_{m}=\left[\begin{array}{c} t_{m}^{0,0},\ldots ,t_{m}^{0,j},\ldots ,t_{m}^{0,6}\\ \vdots \\ t_{m}^{i,0},\ldots ,t_{m}^{i,j},\ldots ,t_{m}^{i,6}\\ \vdots \\ t_{m}^{53,0},\ldots ,t_{m}^{53,j},\ldots ,t_{m}^{53,6} \end{array}\right]$$
(8)$$\tilde{T_{m}}=\left[\begin{array}{c} \tilde{t_{m}^{0,0}},\ldots ,\tilde{t_{m}^{0,j}},\ldots ,\tilde{t_{m}^{0,6}}\\ \vdots \\ \tilde{t_{m}^{i,0}},\ldots ,\tilde{t_{m}^{i,j}},\ldots ,\tilde{t_{m}^{i,6}}\\ \vdots \\ \tilde{t_{m}^{53,0}},\ldots ,\tilde{t_{m}^{53,j}},\ldots ,\tilde{t_{m}^{53,6}} \end{array}\right]$$

The input NIPI, AEA and CDEA indicators for seven weeks are extracted by CNN and the hidden layer vector $I_{n}$ is obtained, as shown in Formula (9), where $n_{t}$, $a_{t}$, $c_{t}$ are the feature vector of NIPI, AEA, CDEA in seven weeks, respectively.
(9)$$I_{n}=\left[n_{1},\ldots ,n_{t}\;,a_{1},\ldots ,a_{t},c_{1},\ldots ,c_{t}\right]$$

According to expert consultation, set the thresholds of $\tilde{P_{m}}$ and $\tilde{T_{m}}$ are set, which are $\delta_{p}$ and $\delta_{t}$, respectively. C represents the importance sequence. Then we get the attention score in Formula (11), where W is the learnable parameter matrix, and $k_{i}$ is the sequence of $I_{n}$.
(10)$$C=\overset{i+7}{\underset{i}{\sum }}\overset{6}{\underset{j=0}{\sum }}\left[1\;\mathrm{or}\;0\;\mathrm{if}\;\tilde{p_{m}^{i,j}}>\delta_{p}\right]\mathrm{or}\left[1\;\mathrm{or}\;0\;\mathrm{if}\;\tilde{t_{m}^{i,j}}>\delta_{t}\right]$$
(11)$$e=I_{n}{}^{T}\tan h\left(W\left[C;k_{i}\right]\right)$$

Softmax function is used to normalize the attention score and get the weight of each feature vector in Formula (12).
(12)$$a=softmax\left(e\right)=\frac{\exp\left(e\right)}{\sum \exp\left(e\right)}$$

Finally, the model combines the calculated attention influence vector with the CNN hidden layer vector to optimize the model extraction result, as shown in Formula (13).
(13)$$Attention\left(NIPI,AEA,CDEA\right)=\sum \;a\cdot I_{n}$$

## 3. Results

### 3.1. Data Set and Experimental Parameters

#### 3.1.1. Data Set

In this paper, security risk assessment indexes for each vegetable in each province are predicted, and the total length of time series for each vegetable in each province is 53 weeks in the experiment. The pre-processed data set is divided into a training set, validation set and test set, and the ratio is 6:3:1 according to the number of the sample data set.

#### 3.1.2. Experimental Environment

For the experimental environment, the operating system was a 64-bit Windows 10 operating system, the processor was Intel CORE i7-9700F@3.00GHz eight-core, the memory was 16 GB, and the graphics card was Nvidia GeForce RTX3060.

An open-source deep learning framework based on PyTorch (https://pytorch.org/ (accessed on 26 February 2022)) is used to construct a deep learning model for experimental platform development.

#### 3.1.3. Experimental Parameters

The attention mechanism-based CNN-AOA-LSTM model is composed of two convolutional layers, two pooling layers and the full connection layer, and Relu, as shown in Formula (14), is the activation function.
(14)$$f\left(x\right)=\max\left(0,W^{T}x+b\right)$$
where W represents the weight vector, b represents the bias vector, and x represents the input vector, which comes from the output vector of the neural network at the upper layer. W and b are network parameters that can be learned.

The first convolutional layer has 64 convolutional kernels, and the size is set to 1 × 4. The second convolutional layer has 32 convolutional kernels with a size of 1 × 3 and a step size of 2. The pooling layer selects the maximum pooling mode, and then connects to the full connection layer for transformation and output. The AOA-LSTM network contains two hidden layers with 30 and 20 cells, respectively.

### 3.2. Model Evaluation Indexes

The prediction of the security risk level is affected by the combination of the above three evaluation indexes; therefore, it is necessary to evaluate the performance of the single evaluation index of the three indexes and the accuracy of the security risk level determined by the indexes.

#### 3.2.1. Prediction Performance Evaluation Indexes

This paper adopts root mean square error (RMSE) and mean absolute error (MAE) to evaluate the predictive effectiveness of the NIDI, AEA and CDEA evaluation indexes in the proposed food security risk level prediction model. The calculation method of these two indexes is as follows:(15)$$\mathrm{RMSE}=\sqrt{\frac{1}{n}\overset{n}{\underset{i=1}{\sum }}\tilde{(x_{i}}-x_{i}{)}^{2}}$$
(16)$$\mathrm{MAE}=\frac{1}{n}\overset{n}{\underset{i=1}{\sum }}\left|\tilde{x_{i}}-x_{i}\right|$$
where $x_{i}$ is the actual value of the evaluation index of the week, and $\tilde{x_{i}}$ is the predicted value of the risk evaluation index of the week.

#### 3.2.2. Prediction Accuracy Evaluation Index

In this paper, three evaluation indexes are adopted: precision, recall rate and F1. Specific calculation methods of various indexes are as follows:(17)$$P_{i}=\frac{TP_{i}}{TP_{i}+FP_{i}}$$
(18)$$R_{i}=\frac{TP_{i}}{TP_{i}+FN_{i}}$$
(19)$$F1=\frac{2*P*R}{P+R}$$

In the precision calculation Formula (17) (precision hereinafter referred to as *P*), $TP_{i}$ represents that the model predicts the number of the positive class as positive classes, and $FP_{i}$. represents the model predicts the number of the negative class as positive classes.

In the Formula (18) for recall (recall hereinafter referred to as *R*), $TP_{i}$ represents the that the model predicts the number of the positive class as positive classes, and $FN_{i}$ that the model predicts the number of the positive class as negative classes.

The data set used in this paper is a balanced data set. Because precision and recall are a pair of contradictory quantities when *P* is high, *R* tends to be relatively low, and when *R* is high, *P* tends to be relatively low, so to better evaluate the performance of the classifier, generally use F1 score as an evaluation criterion to measure the comprehensive performance of the classifier

### 3.3. Security Risk Assessment and Classification

#### 3.3.1. Security Risk Assessment Indexes

In order to comprehensively evaluate the hazard of carbofuran in vegetables, first we calculated the weekly NIPI, AEA and CDEA values of each vegetable in each province from January to December 2019 by the analytic hierarchy process (AHP). Taking Beijing as an example, the data set of security risk assessment indexes for eight vegetables is shown in Figure 3.

#### 3.3.2. Security Risk Classification

After obtaining the food security risk assessment indexes, it can be seen that different indexes differ greatly in order of magnitude. In order to avoid the impact of the assessment effect caused by the neglect of indexes with a smaller order of magnitude, data normalization is necessary [36]. In this study, NIPI, AEA and CDEA are selected as features based on the k-means++ clustering algorithm.

Silhouette coefficient is a means of evaluating the clustering effect. It was first proposed by J. [37] in 1986. It can be used to evaluate the influence of different algorithms or different operating modes of algorithms on clustering results on the basis of the same original data. In this paper, silhouette coefficients are used to measure how many clustering categories are best. Figure 4 shows the fractions of silhouette coefficients for the clustering categories from 3 to 7.

As shown in Figure 4, the silhouette coefficient of category 5 in the clustering result is the largest, indicating that the instances in the cluster are compact and the distance between clusters is large. Therefore, the normalized data set is divided into five categories by the k-means++ algorithm, and the indexes of each cluster center are shown in Table 1. The distance between the cluster center and the origin is calculated according to the normalized index, and the security risk levels of categories 1–5 are defined as low–high, respectively.

The clustering results of security risk levels based on the k-means++ algorithm are shown in Figure 5. Among them, NIPI, AEA and CDEA are represented by three-dimensional system of coordinate, and security risk level is represented by color. In the following, future security risk assessment indexes will be divided into specific security risk levels based on clustering centers.

#### 3.3.3. Analysis of Security Risk Classification Results

The statistics of security risk assessment indexes of the five clustering centers are shown in Figure 6. The three clustering indexes increase successively with the increase of security risk level, and the indexes of categories 1 and 2 are much smaller than those of categories 4 and 5. It can be seen that the security risk level of categories 1 and 2 is relatively low, while that of categories 4 and 5 is relatively high. The distribution of security risk levels is shown in Figure 7. Vegetables with low and medium-low security risk levels account for 88.6% of the total risk, and vegetables with medium-high and high security risk levels account for 6.0%.

### 3.4. Security Risk Level Prediction Model of CNN-AOA-LSTM Based on Attention Mechanism

In order to prove the effectiveness of the CNN-AOA-LSTM model based on the attention mechanism, a series of comparative experiments are carried out in this paper. At present, LSTM, GRU and RNN are the best neural network models for time series analysis; therefore, this paper adopts LSTM, GRU and RNN models as comparison models. Food security risk assessment indexes of each vegetable in each province are predicted, respectively, and then distinguish the security risk level. The experimental results of several common models are given and compared with the prediction model proposed in this paper.

Similarly, taking Beijing as an example, Figure 8, Figure 9, Figure 10, Figure 11, Figure 12, Figure 13, Figure 14 and Figure 15 shows food security risk assessment indexes of eight kinds of vegetables predicted by four models, respectively. In this paper, the prediction step is 7, so the first seven weeks are a window period, and the prediction is not made in 0 to 6 weeks. The prediction starts at week 7. The 0~46 weeks shown in the figure are the weeks that can be predicted, and the actual 7~53 weeks. In the figure, 0~39 weeks are the training set, and 40–46 weeks are the test set. As it is necessary to predict the evaluation indexes of 20 provinces, RMSE and MAE are used to conduct statistical analysis on the three evaluation indexes of 20 provinces predicted by the four models. As can be seen from the figure, most of the predicted curves are consistent with the actual curves in the process of prediction. However, the predicted curve and the actual curve of some vegetables deviate significantly. This paper conducts research and analysis on such situations, and finds that the serious deviation is mainly due to the change of supply chain caused by prominent events. For example, eight batches of unqualified food were exposed in Shandong, involving vegetables and so on, which flowed into Beijing in November 2019.

Figure 16 and Figure 17, respectively, show RMSE and MAE of three indexes of eight vegetables, predicted by four models. As can be seen from the experimental results, the prediction model proposed in this paper has the minimum RMSE and MAE values for eight kinds of vegetables, which is superior to other models. In terms of the NIPI index, the predicted values of RNN model and LSTM model on four kinds of vegetables deviate greatly from the correct values. In terms of AEA and CDEA indexes, the prediction effect of the four models on solanaceous vegetables is slightly lower, but the index prediction of other vegetables is very close to the actual value. In general, the RNN model has the worst prediction effect on all indexes.

After predicting NIPI, AEA and CDEA evaluation indexes, the distance between risk evaluation indexes and security risk levels is calculated, and the security risk level is determined. The average accuracy of the prediction results of the four models is statistically analyzed, as shown in Table 2.

Experimental results show that the CNN-AOA-LSTM prediction model based on the attention mechanism is significantly better than the other three models in terms of precision and recall rate, and has the best overall performance in F1. Since the small sample data set constructed independently is adopted in this paper, large-scale training cannot be carried out in prediction. Therefore, attention mechanism is introduced when features are extracted through the CNN network model. When the extracted features enter the LSTM network model, the AOA algorithm is introduced to optimize the LSTM model to optimize the model’s hyperparameters and improve the prediction accuracy of evaluation indexes.

In order to verify the reliability of deep learning in pesticide residue prediction, we randomly select the high-risk vegetables in a province, such as leeks in the Zhengzhou Economic Development Zone, Henan province, on 10 December 2019, in which carbofuran was over the standard. Subsequently, we find that the indicators of carbofuran in bulb vegetables in this province were also in the middle and high grades in 43rd week, as shown in Figure 18. It shows that the prediction results of deep learning are reliable when there is no emergency in the supply chain.

## 4. Discussion

In order to supervise the high-risk areas of dietary intake of carbofuran pesticide residues and effectively guarantee the food quality and security of residents, this paper establishes the prediction model of security risk level of carbofuran pesticide residues in Chinese vegetables based on deep learning. The prediction is carried out based on a small sample data set. The experimental results show that the prediction precision of CNN-AOA-LSTM based on the attention mechanism proposed in this paper reaches 93.37%, which meets the risk management requirements of weekly food sampling reports. At the same time, security risk classification and prediction based on systematic assessment can make the food supervision department objectively determine the key supervision of provinces and vegetable types, so as to strengthen the early control of food security risks, reduce the cost of risk management, provide safe and assured food for consumers, protect the interests of consumers, and maintain public health and security. The proposed approaches in the paper can combine other identification algorithms to study the computer vision problems [38,39,40,41] and can be applied to other fields such as prediction processing and engineering application systems [42,43,44,45,46].

## Figures and Tables

**Figure 1 foods-11-01061-f001:**
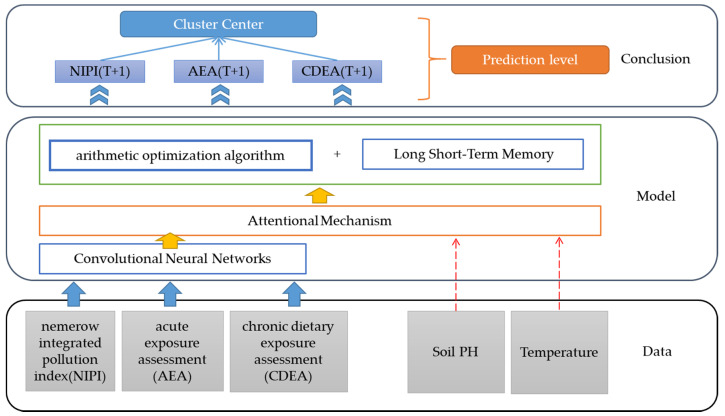
CNN-AOA-LSTM food security risk level prediction model based on the attention mechanism.

**Figure 2 foods-11-01061-f002:**
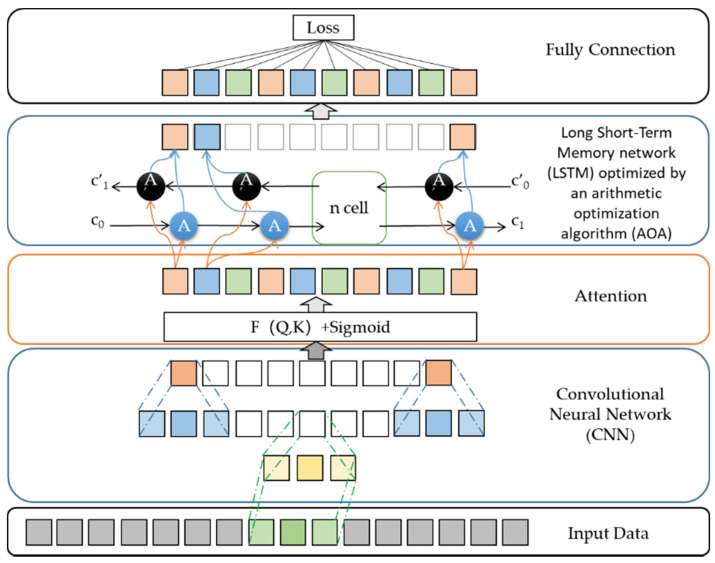
The structure of the neural network model.

**Figure 3 foods-11-01061-f003:**
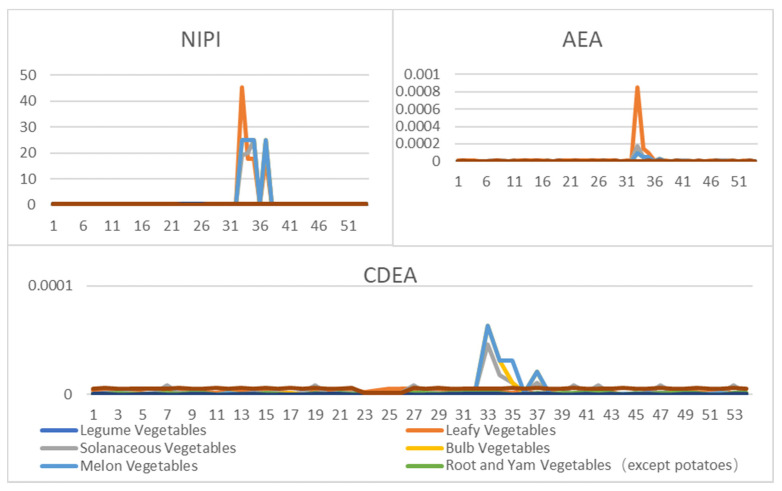
Food security risk assessment indexes of eight kinds of vegetables per week in Beijing from January to December.

**Figure 4 foods-11-01061-f004:**
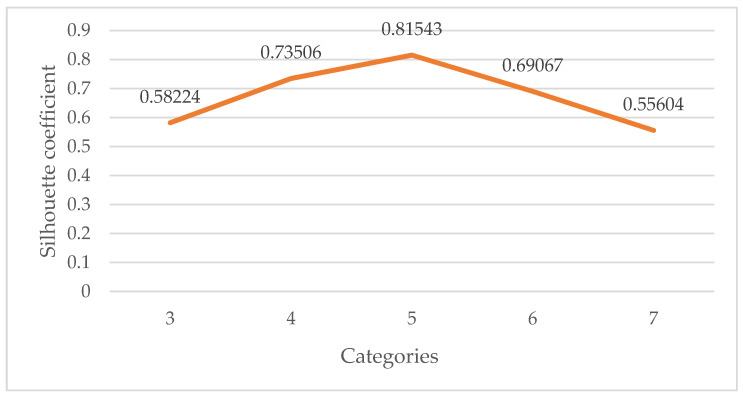
Silhouette coefficients for the clustering categories from 3 to 7.

**Figure 5 foods-11-01061-f005:**
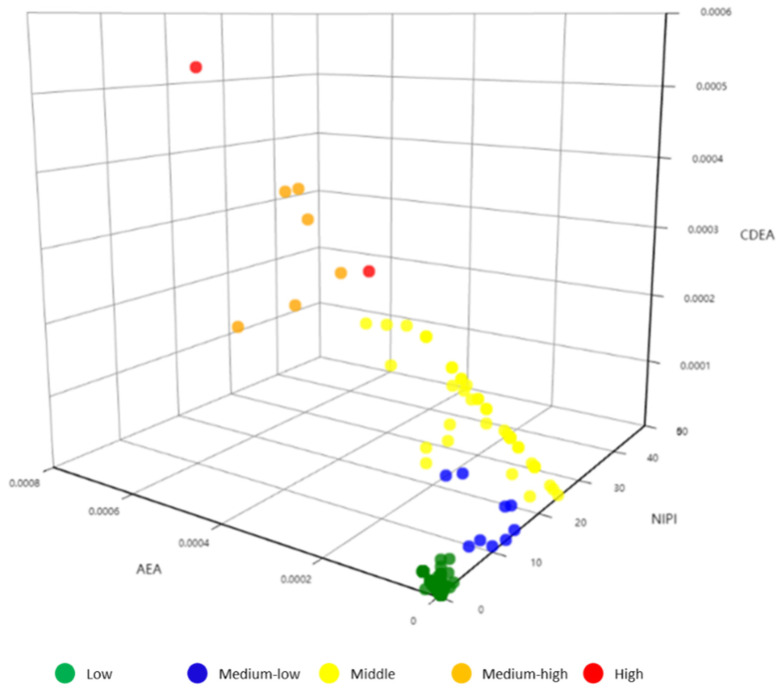
K-means++ clustering results.

**Figure 6 foods-11-01061-f006:**
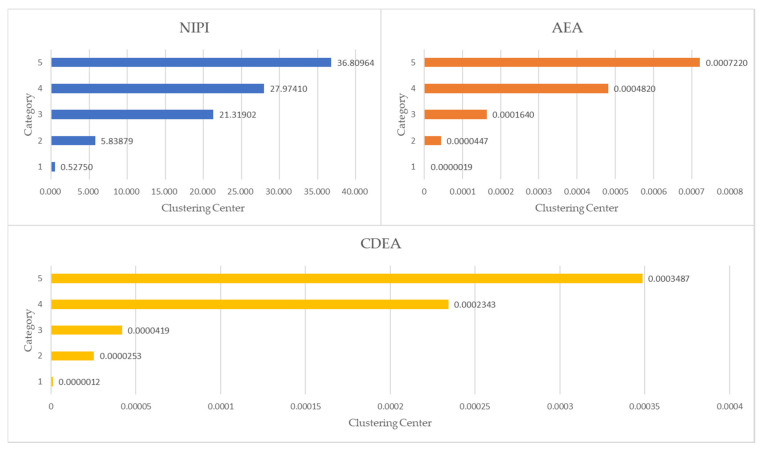
Distribution of different indexes in five cluster centers.

**Figure 7 foods-11-01061-f007:**
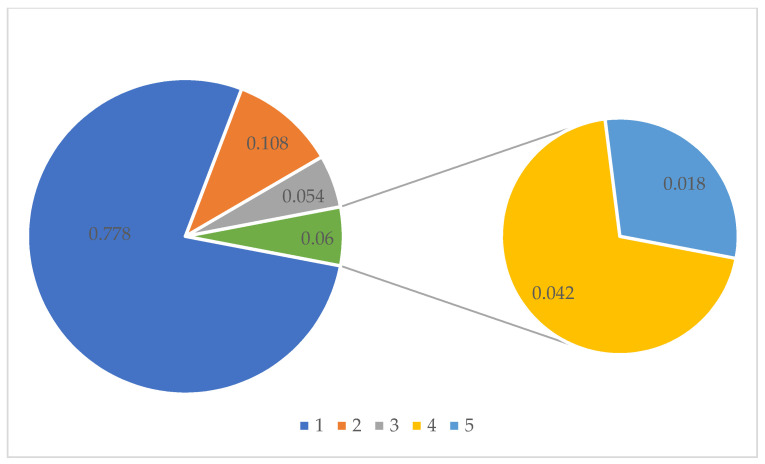
The proportion of five security risk levels.

**Figure 8 foods-11-01061-f008:**
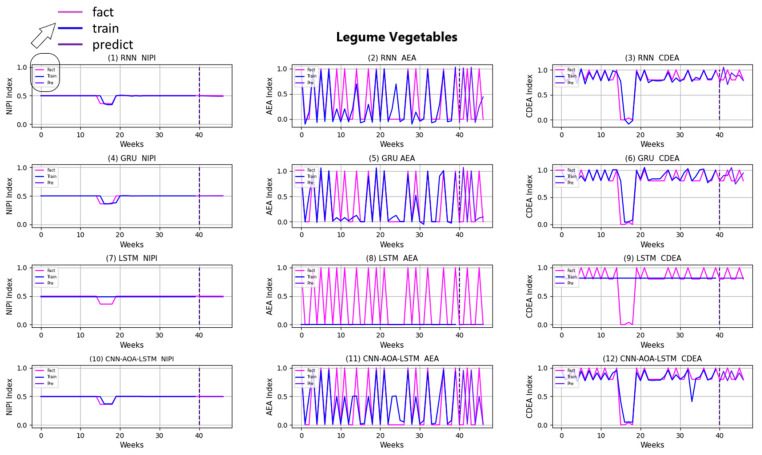
Prediction results of indexes of legume vegetables.

**Figure 9 foods-11-01061-f009:**
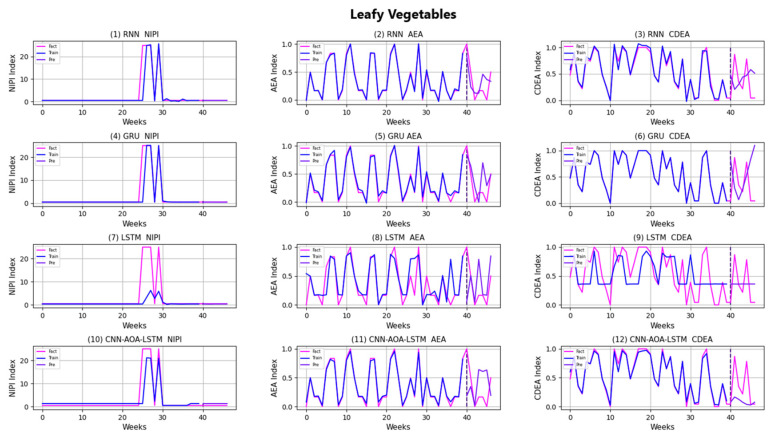
Prediction results of indexes of leafy vegetables.

**Figure 10 foods-11-01061-f010:**
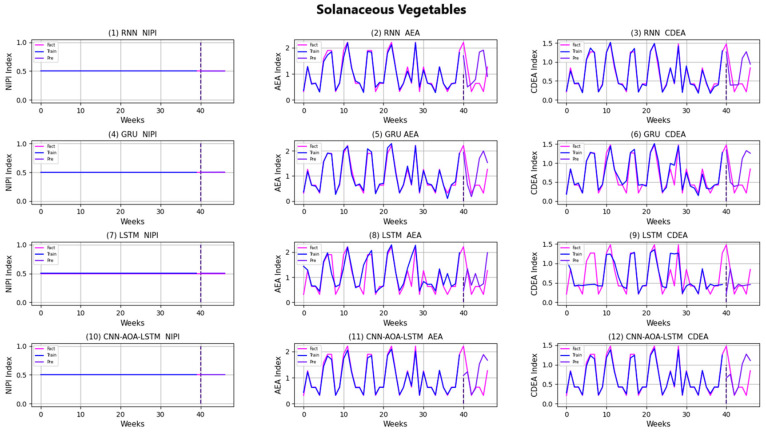
Prediction results of indexes of solanaceous vegetables.

**Figure 11 foods-11-01061-f011:**
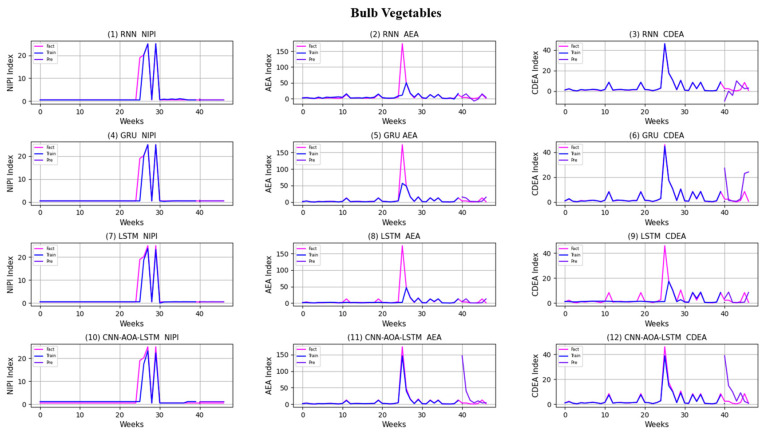
Prediction results of indexes of bulb vegetables.

**Figure 12 foods-11-01061-f012:**
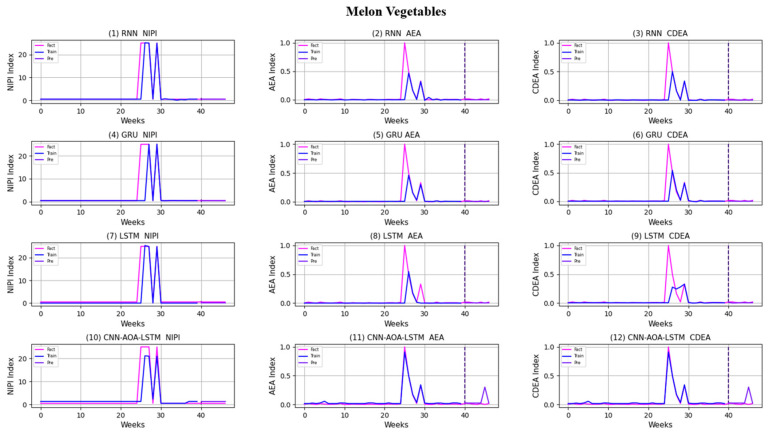
Prediction results of indexes of melon vegetables.

**Figure 13 foods-11-01061-f013:**
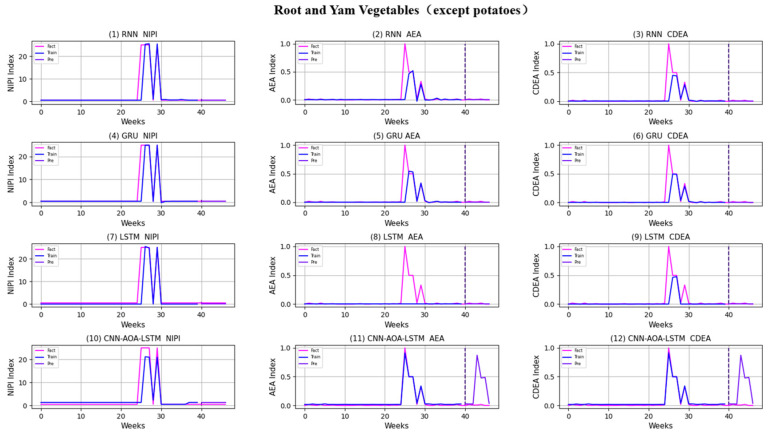
Prediction results of indexes of root and yam vegetables (except potatoes).

**Figure 14 foods-11-01061-f014:**
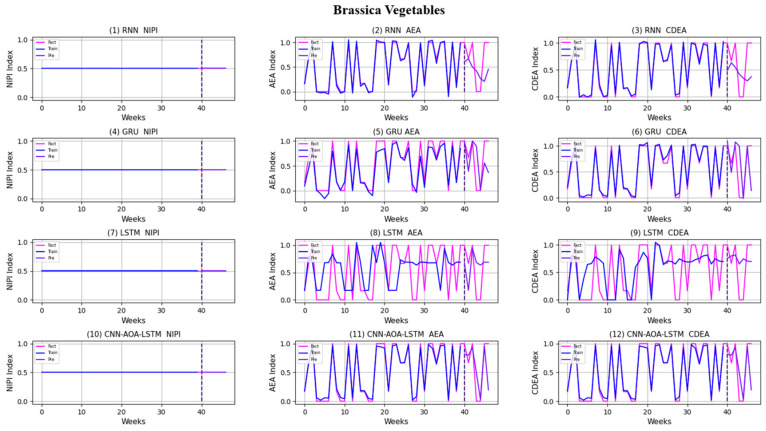
Prediction results of indexes of brassica vegetables.

**Figure 15 foods-11-01061-f015:**
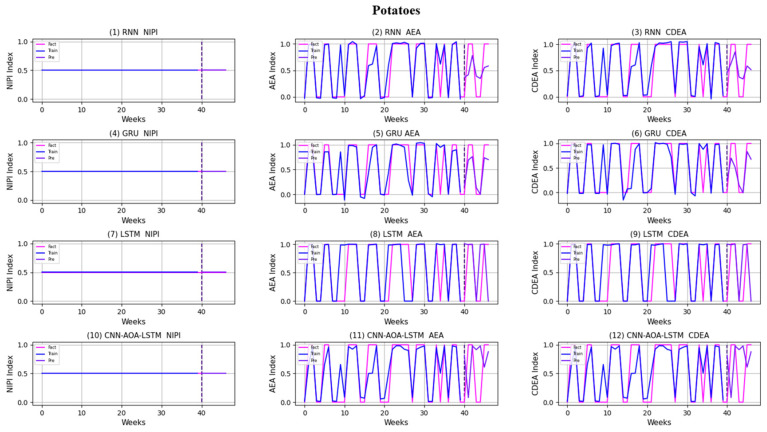
Prediction results of indexes of potatoes.

**Figure 16 foods-11-01061-f016:**
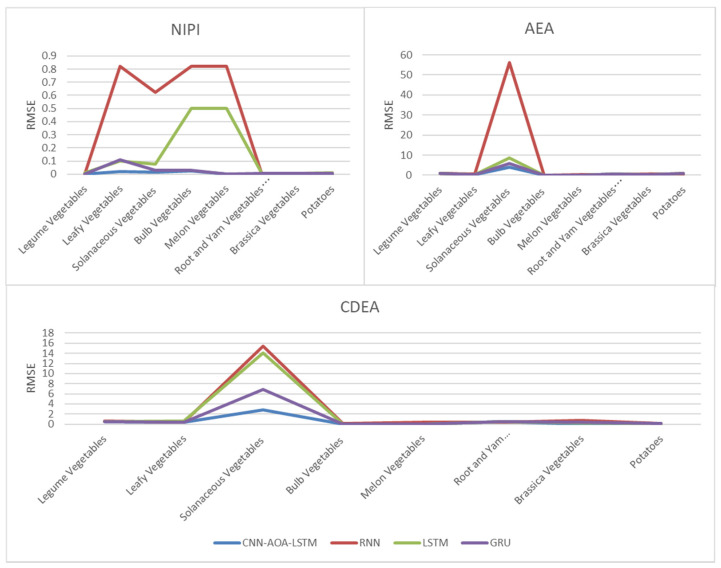
RMSE of NIPI, AEA and CDEA indexes.

**Figure 17 foods-11-01061-f017:**
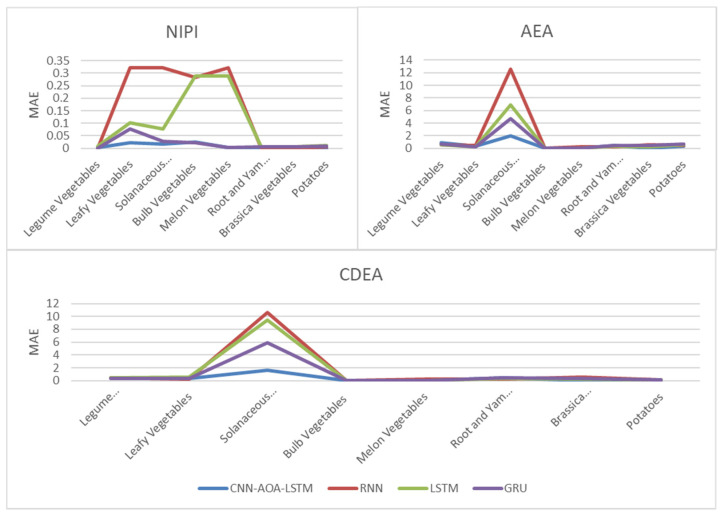
MAE of NIPI, AEA and CDEA indexes.

**Figure 18 foods-11-01061-f018:**
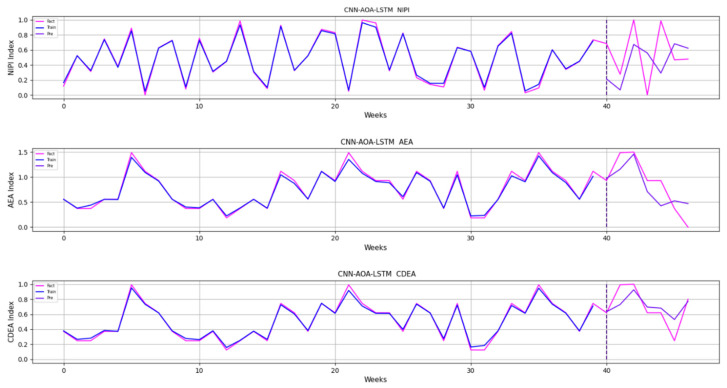
Random sample analysis of predicted results.

**Table 1 foods-11-01061-t001:** Security risk assessment indexes of five cluster centers.

Category	NIPI	AEA	CDEA	Risk Level
1	0.52750	0.0000019	0.0000012	Low
2	5.83879	0.0000447	0.0000253	Medium-low
3	21.31902	0.0001640	0.0000419	Middle
4	27.97410	0.0004820	0.0002343	Medium-high
5	36.80964	0.0007220	0.0003487	High

**Table 2 foods-11-01061-t002:** Accuracy evaluation index of security risk level prediction.

Model	Index-Data		
	P%	R%	F1%
RNN	74.38	73.69	74.03
LSTM	79.37	78.73	79.05
GRU	87.25	86.51	86.88
CNN-AOA-LSTM	93.37	93.12	93.24

## Data Availability

Restrictions apply to the availability of these data. Data was obtained from the State Administration for Market Regulation Statistics and are available at http://spcj.gsxt.gov.cn with the permission of the State Administration for Market Regulation Statistics.

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
