# Peer review of "Security Risk Level Prediction of Carbofuran Pesticide Residues in Chinese Vegetables Based on Deep Learning"

_foods, 2022, doi:10.3390/foods11071061_

Round 1
Reviewer 1 Report
The work is really interesting but I think the text needs to be worked on a bit.
Errors throughout the text
Check the inconsistent spacings, e.g.
. Then the evaluation index value of the security risk ... or
. The prediction model proposed in this paper provides
Leave only the last name in the citations e.g
Weizhong He et al. [8] or Liping Fang et al. [10] or Kumari D et al. [
abbreviations need to be introduced, preferably the first time you use them in the text.
Abstract
Experimental results show that the accuracy of the model proposed CNN-AOA-LSTM (what do these abbreviations mean? CNN? AOA? LSTM?) prediction model based on attention mechanism is 6.12% to 18.99% higher than several commonly used deep neural network models (RNN, LSTM, and GRU ). if possible use the full term instead of the abbreviation.
1. Introduction
W P Yu et al. [16] used ARIMA (Autoregressive Integrated Moving Average Model) to predict pesticide residues in two vegetables in different seasons in Sichuan, China [16]. Zhou Xin et al [17] predicted the heavy metal content in lettuce leaves by deep learning combined with hyperspectral imaging technology [17].
In conclusion, tThis paper establishes a deep learning-based prediction model of security risk level of carbofuran pesticide residue in Chinese vegetables by analyzing 200,000 national monitoring sampling data of carbofuran pesticide residue contamination in vegetables in 2019.
Then, the data sources are clustered by K-Means++ algorithm (add reference) to classify...
The last paragraph is a bit confusing. Try to order the sentences to clarify the idea.
Materials and Methods
.1.1. Data Sources.
How many features did the data set initially have?
Could you mention some beside the ones used?
2.4 CNN-AOA-LSTM security risk level prediction model based on attention mechanism
Please explain models in a bit more depth. add references.
describe the RELU.
In what environment were the algorithms implemented?
What other software was used?
Secondly, this paper introduces the attention mechanism to allocate the probability weight to enhance the proportion of useful information.
Delve into the attention mechanism. add references.
Introduce the Silhouette coefficients. Add reference.
Last but not least add a paragraph explaining that Long Short-Term Memory (LSTM), Gated Recurrent Unit (GRU), and Recurrent Neural Networks (RNN) were used as a control to evaluate the effectiveness of the proposed model.
3. Results
Table 2. Experimental results of security risk level precision
Add table captions, describe P%, R% and F1%
Figure 7- delete the sentence Percentage of ...
Similarly, taking Beijing as an example, Figure 8-15 shows food security risk assessment indexes of 8 kinds of vegetables predicted by four models respectively. The prediction week starts from the 40th week and the step length is 7 ( and ends?)
Could you explain why the prediction curve sometimes deviates significantly from the actual curve?
Reviewer 2 Report
This is a very interesting paper and presents a unique approach to security risk level of carbofuran pesticide residues in Chinese Vegetables. The information presented will be of interest to government. However, the paper suffers for some limits:
1. AOA model is not as common as CNN and LSTM, this paper should describe AOA model in more detail, especially the mechanism of AOA optimizing LSTM parameters.
2. Formula 5 is missing the left parenthesis.
3. References No. 31 and 32 are missing in this paper, especially reference 31 should provide details of AOA model.
Reviewer 3 Report
I have only one major comment for the authors.
Whether the risk predicted using deep learning is compared with actual inicidence of the pesticide residues?
To understand the reliability of DL in pesticide residues, it is better to analyse the samples from some provinces randomly.
